

# Identification of research trends concerning application of stent implantation in the treatment of pancreatic diseases by quantitative and biclustering analysis: a bibliometric analysis

Xuan Zhu[1,2], Xing Niu[3], Tao Li[4], Chang Liu[5], Lijie Chen[6] and Guang Tan[7]

[1] Institute of Translational Medicine, China Medical University, Shenyang, Liaoning, China

[2] Department of General Surgery, Anshan Hospital, First Affiliated Hospital of China Medical University, Anshan, Liaoning, China

[3] Department of Second Clinical College, Shengjing Hospital affiliated to China Medical University, Shenyang, Liaoning, China

[4] Department of General Surgery, Fushun Mining Bureau General Hospital, Fushun, Liaoning, China

[5] Department of General Surgery, First Affiliated Hospital of China Medical University, Shenyang, Liaoning, China

[6] Department of Third Clinical College, China Medical University, Shenyang, Liaoning, China

[7] Department of Hepatobiliary Surgery, First Affiliated Hospital of Dalian Medical University, Dalian, Liaoning, China

Corresponding author
Xuan Zhu, xzhu@cmu.edu.cn

## ABSTRACT

**Objectives.** In recent years, with the development of biological materials, the types and clinical applications of stents have been increasing in pancreatic diseases. However, relevant problems are also constantly emerging. Our purpose was to summarize current hotspots and explore potential topics in the fields of the application of stent implantation in the treatment of pancreatic diseases for future scientific research.

**Methods.** Publications on the application of stents in pancreatic diseases were retrieved from PubMed without language limits. High-frequency Medical Subject Headings (MeSH) terms were identified through Bibliographic Item Co-Occurrence Matrix Builder (BICOMB). Biclustering analysis results were visualized utilizing the gCLUTO software. Finally, we plotted a strategic diagram.

**Results.** A total of 4,087 relevant publications were obtained from PubMed until May 15th, 2018. Eighty-three high-frequency MeSH terms were identified. Biclustering analysis revealed that these high-frequency MeSH terms were classified into eight clusters. After calculating the density and concentricity of each cluster, strategy diagram was presented. The cluster 5 ''complications such as pancreatitis associated with stent implantation'' was located at the fourth quadrant with high centricity and low density.

**Conclusions.** In our study, we found eight topics concerning the application of stent implantation in the treatment of pancreatic diseases. How to reduce the incidence of postoperative complications and improve the prognosis of patients with pancreatic diseases by stent implantation could become potential hotspots in the future research.

## INTRODUCTION

In recent years, stents play an increasingly essential part in pancreatic diseases such as plastic stents, self-expanding metal stents, biodegradable stents, radioactive particle stents and so on. As an example, the covered metal stents reduce the incidence of complications of biliary obstruction caused by pancreatic cancer. It has been reported that percutaneous insertion of short metal stents supplies for a secure treatment, which is beneficial for patients in resectable pancreatic head cancer with jaundice (*Briggs et al., 2010*). With in-depth research, an irradiation pancreatic stent may provide longer patency and better patient survival (*Zhu et al., 2018*). And endoscopic application significantly improves the therapeutic effect of pancreatic stent (*Baron, 2014*). Pancreatic cancer is a common digestive system cancer with high mortality. And the 5-year survival rate has increased from 3% to 8% over the past decade years (*Torre et al., 2015*; *Siegel, Miller & Jemal, 2018*). So far, surgical resection is the only possible treatment option. However, postoperative complications worsen the patient's prognosis and have been one of the leading causes of death after surgery. Multiple plastic stents or covered self-expandable metallic stent could relief bile duct stricture caused by chronic pancreatitis (*Haapamäki et al., 2015*).

There have been few studies on the application of stents in pancreatic diseases by use of bibliometrics. Bibliometric method, as a quantitative analysis method, is used to determine the evolution of science exploration over the past decade years (*Su & Lee, 2010*; *Thompson & Walker, 2015*). Co-word analysis is an important scientometric method for identifying research hotspots in a certain field. Co-word analysis was proposed by French bibliographers in the 1970s. Its principle is mainly to count the frequency of simultaneous occurrence of words in the literature. The clustering analysis, association analysis, multi-dimensional scale analysis and other methods are utilized to analyze the relationship between words (*Yao et al., 2014*). Therefore, co-word analysis can be used to outline the current state of literature research in a field and to predict the future trends (*Hong et al., 2016*). Co-word analysis method reveals the intricate relationships between many objects in an intuitive way such as numerical values and graphics. Therefore, it can avoid the subjective problems brought by the previous reviews which were summarized by authors. Cluster analysis can be used to obtain semantic relationships for research topics (*Cheng & Church, 2000*). In our study, we made double-clustering analysis, which can cluster the rows and columns of a matrix simultaneously (*Hartigan, 1978*). Therefore, it can easily cluster global information and analyze high-dimensional data. The strategic diagram is used to describe the internal contact situation and the interaction between the fields in a research field based on the co-word matrix and clustering analysis, and further analyze the development of research hotspots in a certain subject. The strategic diagram displays the positional relationship of the clusters in the plane coordinates in a visual form. The quadrant structure and changes

of the research subject are described according to the position and variation of the quadrant of the cluster.

Hence, we constructed a bibliometric analysis by co-word analysis and visualization concerning the application of stents in pancreatic diseases. And strategic diagram was established to explore the development status.

## MATERIALS AND METHODS

### Data obtaining

All publications came from PubMed without the restrictions of languages. The PubMed database has been used to retrieve data in some of the biomedical research (*Le et al., 2019*; *Le, 2019*). PubMed is chosen not only because of the authority and breadth of the literature, but also the normative nature of the Medical Subject Headings (MeSH) keywords, more importantly. MeSH has been applied to index and catalog articles in PubMed. In our study, we collected literature on the application of stents in pancreatic diseases on May 15th, 2018, in order to ensure more current research results. Our research strategy was as follows: (''stents''[MeSH Terms] OR ''stents''[All Fields] OR ''stent''[All Fields]) AND ((''pancreas''[MeSH Terms] OR ''pancreas''[All Fields] OR ''pancreatic''[All Fields]) OR (''pancreatitis''[MeSH Terms] OR ''pancreatitis''[All Fields])) and ''2018/05/15'' [PDAT]. Publication trends were retrieved from GoPubMed (http://www.gopubmed.org) (*Doms & Schroeder, 2005*).

### Literature screening criteria

If a paper concerning application of stents in pancreatic diseases was an original article, we would accept the literature. Meanwhile, media coverage and science briefings were excluded. Furthermore, two researchers separately examined the papers by title, abstract and full text. One researcher excluded 20 articles, and the other researcher excluded 19 articles. And the agreement was 95%, which suggested a strong correlation (*Mandrekar, 2011*). Finally, title, author, institution, country, publication year and MeSH terms of available articles were saved into a new file in XML.

### Data extraction and analysis

XML file was imported into BICOMB for data extraction (*Dehdarirad, Villarroya & Barrios, 2014*; *Hu & Zhang, 2015*; *Lei et al., 2008*). And authors, journals and the frequency ranking of MeSH terms were determined (*Le & Ou, 2016*; *Le, Ho & Ou, 2019*). According to the H index, the terms were first sorted in descending order of terms. Then the high-frequency major MeSH terms were identified if a term with frequency greater than or equal to its sequence number (h) from the list of high frequency terms, and h was the threshold for intercepting high frequency terms. Then, the relationships between the high-frequency major MeSH terms and the source literature were determined utilizing biclustering analysis. Also, a binary matrix was produced using the source literature set generated by BICOMB and the high-frequency MeSH terms as columns and rows.

## Cluster analysis

Then, double clusters and visual analysis were performed by "gCLUTO" version 1.0 software. "gCLUTO" is a graphical cluster toolkit and graphical front-end of the "CLUTO" data clustering library (*Karypis Lab, 2014*; *Li et al., 2015*). The clustering analysis was employed to assess the high-frequency MeSH terms. The clustering method was used to repeat the bisection, cosine as the similarity function, and I2 as the clustering criterion function. By use of different numbers of clusters, two clusters were performed to differentiate the first-rank number of clusters. And the visualizations of high frequency and high-frequency bifocal results with MeSH article were constructed by use of Alpine and Matrix. By means of the semantic corrections between the MeSH terms and the content of typical articles in every group, the relevant topics on the application of stents in pancreatic diseases were obtained. And we made a visualized matrix biclustering of high-frequent major MeSH terms and PubMed Unique Identifiers (PMIDs) of articles on the application of stents in pancreatic diseases.

## Strategic diagram analysis

A two-dimensional table is depicted by plotting themes based on centricity and density. The *X*-axis stands for centrality, namely the closeness between keywords within this category and those within other categories. It indicates the degree of interaction between a subject area and other subject areas. The *Y*-axis represents density, namely the closeness of the keywords within each category. And it indicates that this category maintains and develops its own capabilities (*Callon, Courtial & Laville, 1991*). The above eight categories were assigned to the four quadrants based on the results of the cluster analysis. In addition, excel was utilized to generate strategic diagram.

## Social network analysis

The high frequency MeSH terms co-occurrence matrix was imported into the Ucinet 6.0 (Analytic Technologies Co., Lexington, KY, USA) software. And the social network analysis method was utilized to analyze the subject and knowledge structure of the application of stents in pancreatic diseases. Then the high-frequency MeSH term network was visualized by NetDraw 2.084 software. The nodes represent MeSH terms, and the links stand for the co-occurrence frequency of these terms. And we measured the degree, betweenness and closeness centralities of every node. At the same time, author relationship network was constructed by above methods.

## RESULTS

### Overall evaluation

Based on GoPubMed, we obtained the literature information according to the search strategy: stents [MeSH] and pancreas [MeSH] or "pancreatic diseases" [MeSH]. Figure 1A depicts the distribution of the publication year of corresponding papers. The first article was published in 1977. As time went by, the volume of publications increased year by year. By 2015, it had a downward trend. Figure 1B shows the volume of paper outputs concerning the application of stents in pancreatic diseases in the first 20 countries. And the

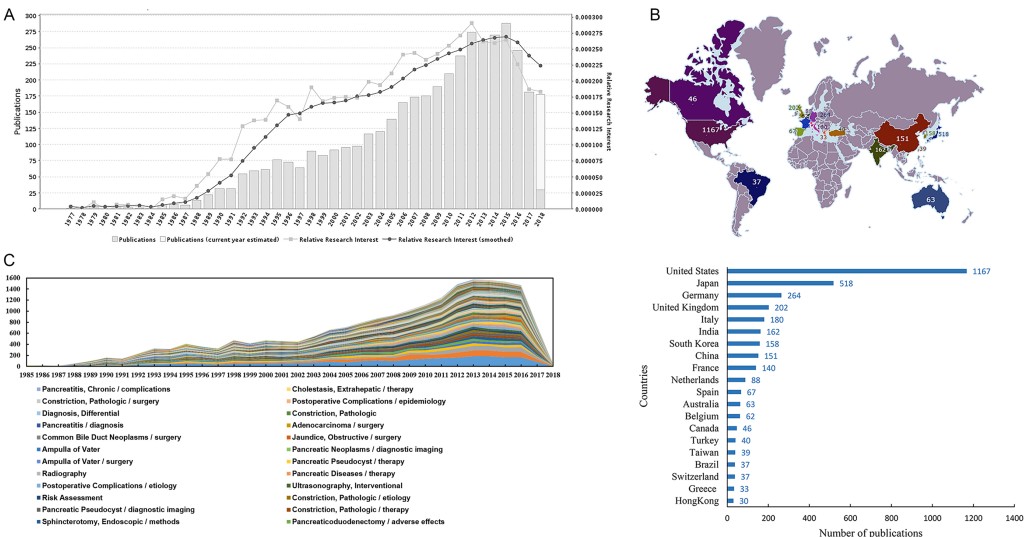

**Figure 1** **The information of literature on the application of stents in pancreatic diseases.** (A) The growth of literature publications about the application of stents in pancreatic diseases from 1977 to 2018. (B) Geographic distribution of research outputs on the application of stents in pancreatic diseases. (C) Annual distribution of MeSH terms about the application of stents in pancreatic diseases.

map was generated by an online website (pixelmap.amcharts.com). The number in the map is the quantity of associated publications for every country or region. The United States stands first with 1,167 publications. Furthermore, we summarized the annual distribution of MeSH terms associated with the application of stents in pancreatic diseases (Fig. 1C). Different colors represent different highly frequent major MeSH terms. We found that these MeSH terms had roughly the same development trend every year from 1985-2018, indicating that they had close associations. As shown in Table 1, the top 29 authors with a cumulative percentage of 27.9483 are listed. "Baron TH" (84, 2.0468%), "Kahaleh M" (81, 1.9737%) and "Isayama H" (65, 1.5838%) are the top three authors. From 1977 to 2018, the 25 most active journals published publications on the application of stents in pancreatic diseases account for 49.92% of all publications. Table 2 demonstrates the 25 most productive journals, as the core journals in the research fields on the application of stents in pancreatic diseases under Bradford's Law. "Gastrointestinal endoscopy", "Endoscopy", "World journal of gastroenterology" are the most active three journals.

## High-frequent major MeSH terms

A total of 4,087 articles were selected until May 15th, 2018. Eighty-three high-frequency MeSH terms were extracted from the listed publications, with a cumulative percentage of 57.5291 (Table 3). "Stents" (2238, 3.8488%), "Treatment Outcome" (1038, 1.7851%) and "Retrospective Studies" (758, 1.3036%) are the top three MeSH terms.

## Cluster analysis

The double cluster analysis results were visualized into mountain visualization and hierarchical cluster tree. In the mountain visualization, the peak and matrix visualizations

**Table 1** The 29 top authors from the listed publications on the application of stents in pancreatic diseases (PubMed sourced until May 2018).

| No. | Author | Frequency | Percentage, %[a] | Cumulative percentage, % |
|-----|--------|-----------|------------------|--------------------------|
| 1 | Baron TH | 84 | 2.0468 | 2.0468 |
| 2 | Kahaleh M | 81 | 1.9737 | 4.0205 |
| 3 | Isayama H | 65 | 1.5838 | 5.6043 |
| 4 | Itoi T | 58 | 1.4133 | 7.0175 |
| 5 | Nakai Y | 50 | 1.2183 | 8.2359 |
| 6 | Varadarajulu S | 49 | 1.194 | 9.4298 |
| 7 | Sherman S | 46 | 1.1209 | 10.5507 |
| 8 | Lehman GA | 41 | 0.999 | 11.5497 |
| 9 | Costamagna G | 39 | 0.9503 | 12.5 |
| 10 | Tada M | 39 | 0.9503 | 13.4503 |
| 11 | Bhasin DK | 38 | 0.9259 | 14.3762 |
| 12 | Koike K | 37 | 0.9016 | 15.2778 |
| 13 | Rana SS | 37 | 0.9016 | 16.1793 |
| 14 | Devière J | 36 | 0.8772 | 17.0565 |
| 15 | Freeman ML | 36 | 0.8772 | 17.9337 |
| 16 | Kogure H | 36 | 0.8772 | 18.8109 |
| 17 | Kozarek RA | 35 | 0.8528 | 19.6637 |
| 18 | Hirano K | 32 | 0.7797 | 20.4435 |
| 19 | Ito K | 31 | 0.7554 | 21.1988 |
| 20 | Wilcox CM | 31 | 0.7554 | 21.9542 |
| 21 | Sasahira N | 31 | 0.7554 | 22.7096 |
| 22 | Sasaki T | 30 | 0.731 | 23.4405 |
| 23 | Huibregtse K | 27 | 0.6579 | 24.0984 |
| 24 | Kim MH | 27 | 0.6579 | 24.7563 |
| 25 | Yamamoto N | 27 | 0.6579 | 25.4142 |
| 26 | Khashab MA | 26 | 0.6335 | 26.0478 |
| 27 | Lee JH | 26 | 0.6335 | 26.6813 |
| 28 | Gupta R | 26 | 0.6335 | 27.3148 |
| 29 | Adler DG | 26 | 0.6335 | 27.9483 |
| | Total | 1,147 | | |

**Notes.**
[a] Proportion of the frequency among 1,147 times' appearance.

express the high-frequency MeSH terms. Each cluster represents a peak marked by cluster number 0–7 in Fig. 2, and the related clusters are described according to the volume, color and height of the peaks. The volume of the peak is directly proportional to the number of MeSH terms in the cluster. Meanwhile, the internal standard deviation of a cluster object is represented by the color of the peak. Blue stands for the high deviation and red represents the low deviation. The peak is the position relative to the other clusters. The closer the distance between the two peaks, the higher the similarity between the two clusters. The height and similarity of each cluster are proportional to each other.

**Table 2** Most active journals on the topic of the application of stents in pancreatic diseases (PubMed sourced until May 2018).

| No. | Top journals | Publications n (%) |
|---|---|---|
| 1 | Gastrointestinal endoscopy | 517 (12.55) |
| 2 | Endoscopy | 339 (8.23) |
| 3 | World journal of gastroenterology | 107 (2.60) |
| 4 | Surgical endoscopy | 101 (2.45) |
| 5 | Digestive endoscopy: official journal of the Japan Gastroenterological Endoscopy Society | 87 (2.11) |
| 6 | The American journal of gastroenterology | 76 (1.85) |
| 7 | Hepato-gastroenterology | 76 (1.85) |
| 8 | Cardiovascular and interventional radiology | 61 (1.48) |
| 9 | Gastrointestinal endoscopy clinics of North America | 61 (1.48) |
| 10 | Digestive diseases and sciences | 59 (1.43) |
| 11 | JOP: Journal of the pancreas | 51 (1.24) |
| 12 | Journal of gastroenterology and hepatology | 51 (1.24) |
| 13 | Journal of vascular and interventional radiology: JVIR | 48 (1.17) |
| 14 | Journal of gastrointestinal surgery: official journal of the Society for Surgery of the Alimentary Tract | 45 (1.09) |
| 15 | Journal of clinical gastroenterology | 45 (1.09) |
| 16 | Pancreas | 44 (1.07) |
| 17 | Pancreatology: official journal of the International Association of Pancreatology (IAP) ... [et al. ] | 40 (0.97) |
| 18 | World journal of gastrointestinal endoscopy | 35 (0.85) |
| 19 | Clinical gastroenterology and hepatology: the official clinical practice journal of the American Gastroenterological Association | 33 (0.80) |
| 20 | Gut | 33 (0.80) |
| 21 | Endoscopic ultrasound | 32 (0.78) |
| 22 | HPB: the official journal of the International Hepato Pancreato Biliary Association | 30 (0.73) |
| 23 | Gan to kagaku ryoho. Cancer & chemotherapy | 29 (0.70) |
| 24 | Journal of hepato-biliary-pancreatic sciences | 29 (0.70) |
| 25 | Annals of surgery | 27 (0.66) |
| | Total | 2056(49.92) |

In Fig. 3, the row labels represent high-frequency MeSH terms, and the PMIDs locate the column labels at the right and bottom of the matrix. The color of each grid suggests the frequency of appearance in a paper. The darker the red, the greater the frequency. Eighty-three high-frequency major MeSH terms are distinguished into eight clusters in matrix visualization. The top and left of the hierarchical tree respectively indicate the relationships among the major MeSH terms and the associations among the papers. Meanwhile, the corresponding article is obviously shown for each high frequency MeSH terms in each cluster.

**Table 3  83 High-frequent major MeSH terms from the listed publications on the application of stents in pancreatic diseases.**

| No. | Major MeSH[a] terms/MeSH subheadings | Frequency, n | Percentage, %[b] | Cumulative percentage, % |
|---|---|---|---|---|
| 1 | Stents | 2238 | 3.8488 | 13.9489 |
| 2 | Treatment Outcome | 1038 | 1.7851 | 27.731 |
| 3 | Retrospective Studies | 758 | 1.3036 | 30.3725 |
| 4 | Cholangiopancreatography, Endoscopic Retrograde | 677 | 1.1643 | 31.5368 |
| 5 | Pancreatic Neoplasms/complications | 544 | 0.9355 | 32.4723 |
| 6 | Follow-Up Studies | 472 | 0.8117 | 33.284 |
| 7 | Drainage/methods | 452 | 0.7773 | 34.0614 |
| 8 | Pancreatic Neoplasms/surgery | 449 | 0.7722 | 34.8335 |
| 9 | Stents/adverse effects | 401 | 0.6896 | 35.5231 |
| 10 | Cholestasis/etiology | 379 | 0.6518 | 36.1749 |
| 11 | Tomography, X-ray Computed | 371 | 0.638 | 36.813 |
| 12 | Pancreatitis/etiology | 338 | 0.5813 | 37.3942 |
| 13 | Cholangiopancreatography, Endoscopic Retrograde/adverse effects | 335 | 0.5761 | 37.9704 |
| 14 | Cholangiopancreatography, Endoscopic Retrograde/methods | 314 | 0.54 | 38.5104 |
| 15 | Prospective Studies | 297 | 0.5108 | 39.0211 |
| 16 | Pancreatic Ducts/surgery | 295 | 0.5073 | 39.5284 |
| 17 | Time Factors | 289 | 0.497 | 40.0255 |
| 18 | Drainage | 281 | 0.4832 | 40.5087 |
| 19 | Palliative Care | 270 | 0.4643 | 40.973 |
| 20 | Endosonography | 254 | 0.4368 | 41.4099 |
| 21 | Cholestasis/therapy | 250 | 0.4299 | 42.2766 |
| 22 | Risk Factors | 244 | 0.4196 | 42.6962 |
| 23 | Pancreatic Neoplasms/pathology | 238 | 0.4093 | 43.1055 |
| 24 | Cholestasis/surgery | 226 | 0.3887 | 43.4942 |
| 25 | Chronic Disease | 198 | 0.3405 | 43.8347 |
| 26 | Drainage/instrumentation | 195 | 0.3354 | 44.17 |
| 27 | Metals | 185 | 0.3182 | 44.4882 |
| 28 | Pancreatic Ducts | 184 | 0.3164 | 44.8046 |
| 29 | Sphincterotomy, Endoscopic | 182 | 0.313 | 45.1176 |
| 30 | Pancreatic Pseudocyst/surgery | 182 | 0.313 | 45.4306 |
| 31 | Recurrence | 179 | 0.3078 | 45.7385 |
| 32 | Pancreatitis/complications | 177 | 0.3044 | 46.0429 |
| 33 | Pancreatitis/surgery | 176 | 0.3027 | 46.3455 |
| 34 | Pancreatic Neoplasms/therapy | 169 | 0.2906 | 46.6362 |
| 35 | Jaundice, Obstructive/etiology | 164 | 0.282 | 46.9182 |
| 36 | Prosthesis Design | 164 | 0.282 | 47.2002 |
| 37 | Pancreatitis/prevention & control | 164 | 0.282 | 47.4823 |
| 38 | Acute Disease | 163 | 0.2803 | 47.7626 |

**Table 3** (*continued*)

| No. | Major MeSH[a] terms/MeSH subheadings | Frequency, n | Percentage, %[b] | Cumulative percentage, % |
|---|---|---|---|---|
| 39 | Equipment Design | 162 | 0.2786 | 48.0412 |
| 40 | Palliative Care/methods | 158 | 0.2717 | 48.3129 |
| 41 | Bile Duct Neoplasms/complications | 157 | 0.27 | 48.5829 |
| 42 | Pancreatitis/therapy | 151 | 0.2597 | 49.1057 |
| 43 | Endoscopy, Digestive System | 150 | 0.258 | 49.3637 |
| 44 | Endoscopy, Digestive System/methods | 146 | 0.2511 | 49.6148 |
| 45 | Survival Rate | 144 | 0.2476 | 49.8624 |
| 46 | Pancreas/surgery | 143 | 0.2459 | 50.1083 |
| 47 | Pancreatic Diseases/surgery | 132 | 0.227 | 50.5692 |
| 48 | Cholangiopancreatography, Endoscopic Retrograde/instrumentation | 131 | 0.2253 | 50.7945 |
| 49 | Pancreaticoduodenectomy | 130 | 0.2236 | 51.0181 |
| 50 | Pancreatic Neoplasms/mortality | 128 | 0.2201 | 51.2382 |
| 51 | Pancreatic Fistula/etiology | 127 | 0.2184 | 51.4566 |
| 52 | Postoperative Complications | 127 | 0.2184 | 51.675 |
| 53 | Endosonography/methods | 125 | 0.215 | 51.89 |
| 54 | Prognosis | 125 | 0.215 | 52.105 |
| 55 | Pancreatic Ducts/pathology | 122 | 0.2098 | 52.3148 |
| 56 | Pancreatic Neoplasms/diagnosis | 121 | 0.2081 | 52.5229 |
| 57 | Endoscopy | 119 | 0.2047 | 52.7275 |
| 58 | Cholestasis, Extrahepatic/etiology | 117 | 0.2012 | 52.9287 |
| 59 | Pancreatic Ducts/diagnostic imaging | 116 | 0.1995 | 53.1282 |
| 60 | Pancreaticoduodenectomy/adverse effects | 115 | 0.1978 | 53.326 |
| 61 | Sphincterotomy, Endoscopic/methods | 114 | 0.1961 | 53.522 |
| 62 | Constriction, Pathologic/therapy | 114 | 0.1961 | 53.7181 |
| 63 | Pancreatic Pseudocyst/diagnostic imaging | 113 | 0.1943 | 53.9124 |
| 64 | Constriction, Pathologic/etiology | 113 | 0.1943 | 54.1068 |
| 65 | Risk Assessment | 113 | 0.1943 | 54.3011 |
| 66 | Ultrasonography, Interventional | 112 | 0.1926 | 54.4937 |
| 67 | Postoperative Complications/etiology | 112 | 0.1926 | 54.6863 |
| 68 | Pancreatic Diseases/therapy | 112 | 0.1926 | 54.8789 |
| 69 | Radiography | 110 | 0.1892 | 55.0681 |
| 70 | Pancreatic Pseudocyst/therapy | 110 | 0.1892 | 55.2573 |
| 71 | Ampulla of Vater/surgery | 110 | 0.1892 | 55.4464 |
| 72 | Pancreatic Neoplasms/diagnostic imaging | 109 | 0.1875 | 55.6339 |
| 73 | Ampulla of Vater | 109 | 0.1875 | 55.8214 |
| 74 | Jaundice, Obstructive/surgery | 108 | 0.1857 | 56.0071 |
| 75 | Common Bile Duct Neoplasms/surgery | 107 | 0.184 | 56.1911 |
| 76 | Adenocarcinoma/surgery | 106 | 0.1823 | 56.3734 |
| 77 | Pancreatitis/diagnosis | 101 | 0.1737 | 56.5471 |
| 78 | Constriction, Pathologic | 101 | 0.1737 | 56.7208 |

**Table 3** (*continued*)

| No. | Major MeSH[a] terms/MeSH subheadings | Frequency, n | Percentage, %[b] | Cumulative percentage, % |
|---|---|---|---|---|
| 79 | Diagnosis, Differential | 97 | 0.1668 | 56.8876 |
| 80 | Postoperative Complications/epidemiology | 95 | 0.1634 | 57.051 |
| 81 | Constriction, Pathologic/surgery | 94 | 0.1617 | 57.2126 |
| 82 | Cholestasis, Extrahepatic/therapy | 92 | 0.1582 | 57.3708 |
| 83 | Pancreatitis, Chronic/complications | 92 | 0.1582 | 57.5291 |

**Notes.**
[a]MeSH: Medical Subject Headings
[b]Proportion of the frequency among 19282 times' appearance.

### Strategic diagram

The centrality and density of the 8 clusters are listed in Table 4. The details of MeSH terms and clusters are shown in Table 5. In Fig. 4, $x$-axis represents the centrality, and $y$-axis stands for the density on the strategy diagram. The four quadrants clockwise from the upper right corner express the first quadrant, the second quadrant, the third quadrant and the fourth quadrant. As shown in Fig. 4A, the clusters in the first quadrant are suggested to be central topics in the network (due to their strong connection with other clusters) and have intense internal relationships (due to high degree of development). The clusters in the second quadrant are peripheral, however, already well-developed topic. The clusters in the third quadrant are both peripheral and undeveloped. The clusters in the fourth quadrant are central and undeveloped, but they are becoming mature to some extent (*Indolfi et al., 2016*).

Figure 4B depicts that cluster 1 and cluster 3 are located in the first quadrant, suggesting that the cluster densities and centrality degrees are all high, that is to say, the MeSH terms in cluster 1 and cluster 3 are closely linked, and research tends to be well-developed. And the orientation is high, indicating that it is at the center of the research network. Cluster 4 and 7 are located in the second quadrant with high density and low centrality, indicating that internal links are close together with a clear topic. The research on this topic is shown to be relatively well-developed, with little correlation with other research. Cluster 0, 2 and 6 are located in the third quadrant, with low density and centrality. MeSH terms of Cluster 0, 2 and 6 are the margins of the entire field. The internal structure is relatively loose and research is yet developed. Cluster 5 is located in the fourth quadrant with low density and high centrality, indicating that it has close relations with other research. However, the research is not found to be well-developed. The research on this topic has potential value, and is now in the exploratory stage; however, more research is required.

### Social network analysis

As shown in Fig. 5A, we constructed the author relationship network. There are 29 nodes which represent 29 authors. The size and location of nodes suggests the decisive role of an author. Links indicate the connection between two authors. In Fig. 5A, the node "Itoi T" was the largest one, which was located in the center of the social network, followed by "Isayama H" and "Sasaki T". Therefore, these authors could play a critical role in the field of the application of stents in pancreatic diseases. Their articles could represent the

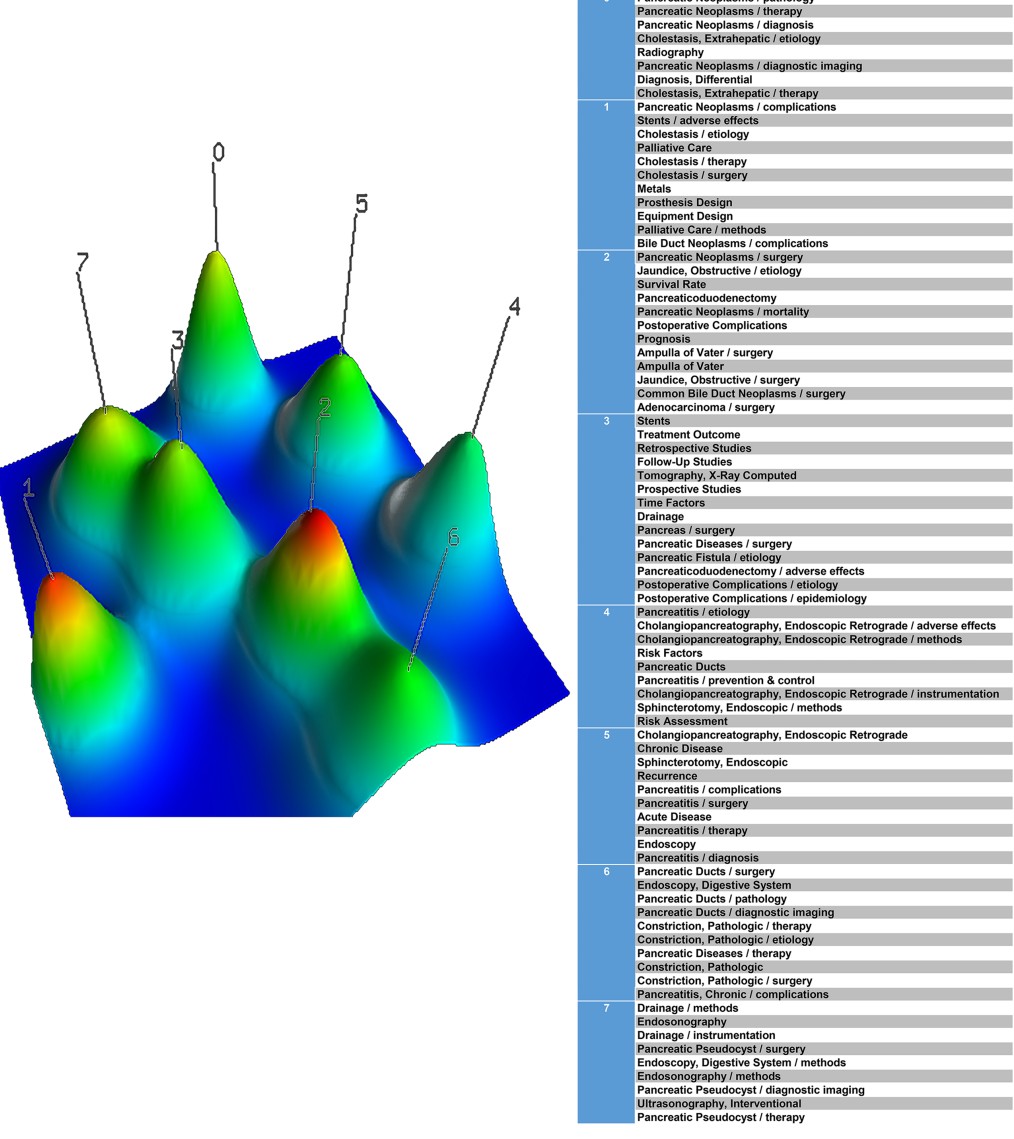

| Cluster | MeSH Terms |
| --- | --- |
| 0 | Pancreatic Neoplasms / pathology |
| | Pancreatic Neoplasms / therapy |
| | Pancreatic Neoplasms / diagnosis |
| | Cholestasis, Extrahepatic / etiology |
| | Radiography |
| | Pancreatic Neoplasms / diagnostic imaging |
| | Diagnosis, Differential |
| | Cholestasis, Extrahepatic / therapy |
| 1 | Pancreatic Neoplasms / complications |
| | Stents / adverse effects |
| | Cholestasis / etiology |
| | Palliative Care |
| | Cholestasis / therapy |
| | Cholestasis / surgery |
| | Metals |
| | Prosthesis Design |
| | Equipment Design |
| | Palliative Care / methods |
| | Bile Duct Neoplasms / complications |
| 2 | Pancreatic Neoplasms / surgery |
| | Jaundice, Obstructive / etiology |
| | Survival Rate |
| | Pancreaticoduodenectomy |
| | Pancreatic Neoplasms / mortality |
| | Postoperative Complications |
| | Prognosis |
| | Ampulla of Vater / surgery |
| | Ampulla of Vater |
| | Jaundice, Obstructive / surgery |
| | Common Bile Duct Neoplasms / surgery |
| | Adenocarcinoma / surgery |
| 3 | Stents |
| | Treatment Outcome |
| | Retrospective Studies |
| | Follow-Up Studies |
| | Tomography, X-Ray Computed |
| | Prospective Studies |
| | Time Factors |
| | Drainage |
| | Pancreas / surgery |
| | Pancreatic Diseases / surgery |
| | Pancreatic Fistula / etiology |
| | Pancreaticoduodenectomy / adverse effects |
| | Postoperative Complications / etiology |
| | Postoperative Complications / epidemiology |
| 4 | Pancreatitis / etiology |
| | Cholangiopancreatography, Endoscopic Retrograde / adverse effects |
| | Cholangiopancreatography, Endoscopic Retrograde / methods |
| | Risk Factors |
| | Pancreatic Ducts |
| | Pancreatitis / prevention & control |
| | Cholangiopancreatography, Endoscopic Retrograde / instrumentation |
| | Sphincterotomy, Endoscopic / methods |
| | Risk Assessment |
| 5 | Cholangiopancreatography, Endoscopic Retrograde |
| | Chronic Disease |
| | Sphincterotomy, Endoscopic |
| | Recurrence |
| | Pancreatitis / complications |
| | Pancreatitis / surgery |
| | Acute Disease |
| | Pancreatitis / therapy |
| | Endoscopy |
| | Pancreatitis / diagnosis |
| 6 | Pancreatic Ducts / surgery |
| | Endoscopy, Digestive System |
| | Pancreatic Ducts / pathology |
| | Pancreatic Ducts / diagnostic imaging |
| | Constriction, Pathologic / therapy |
| | Constriction, Pathologic / etiology |
| | Pancreatic Diseases / therapy |
| | Constriction, Pathologic |
| | Constriction, Pathologic / surgery |
| | Pancreatitis, Chronic / complications |
| 7 | Drainage / methods |
| | Endosonography |
| | Drainage / instrumentation |
| | Pancreatic Pseudocyst / surgery |
| | Endoscopy, Digestive System / methods |
| | Endosonography / methods |
| | Pancreatic Pseudocyst / diagnostic imaging |
| | Ultrasonography, Interventional |
| | Pancreatic Pseudocyst / therapy |

**Figure 2** **A mountain visualization biclustering of 83 high-frequent major MeSH terms and papers on the application of stents in pancreatic diseases.**

maturity of the research area and hot spots. Figure 5B depicts that the network relationships among 83 high-frequent major MeSH terms. The size of nodes suggests the centrality of high-frequent major MeSH terms. In the meanwhile, the thickness of the lines demonstrates the co-occurrence frequency of keywords pairs.

## DISCUSSION

We took advantage of GoPubMed to analyze the publication trends in the field of pancreatic stents. Before 2015, the volume of relevant publications was continuously rising and relative research interest was fluctuating rising. However, beginning with 2015, the volume of
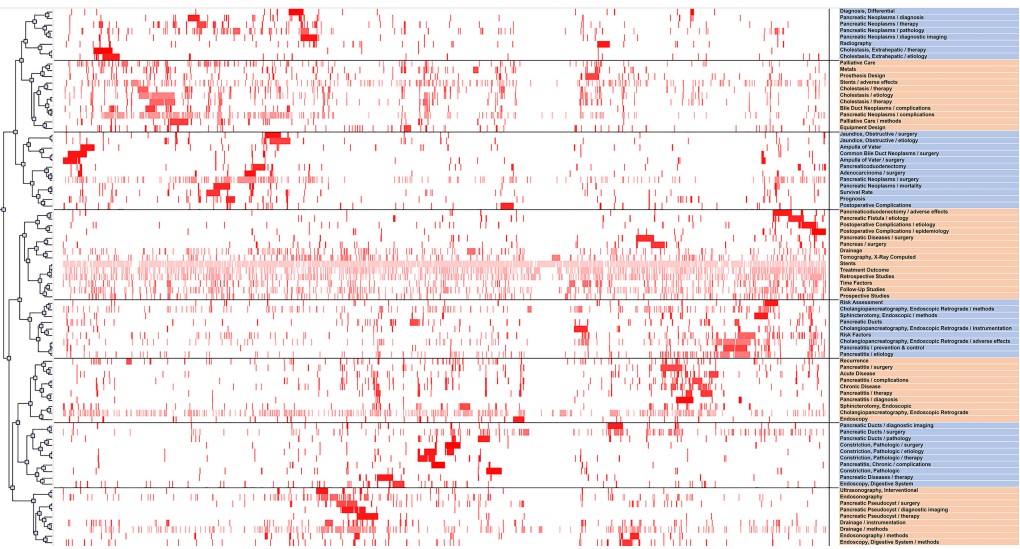

**Figure 3** A visualized matrix biclustering of highly frequent major MeSH terms and PubMed Unique Identifiers (PMIDs) of articles on the application of stents in pancreatic diseases.

**Table 4** The centrality and density of the 8 clusters.

| Cluster | Intra-class link averages | Centrality-X | Intra-class link averages | Density-Y |
|---|---|---|---|---|
| 0 | 8.446666667 | −4.62712 | 33.16071429 | −22.7996 |
| 1 | 15.29292929 | 2.219142 | 75.47272727 | 19.51241 |
| 2 | 9.875586854 | −3.1982 | 31.62878788 | −24.3315 |
| 3 | 24.98033126 | 11.90654 | 98.67032967 | 42.71001 |
| 4 | 12.63963964 | −0.43415 | 68.26388889 | 12.30357 |
| 5 | 13.39589041 | 0.322103 | 55.3 | −0.66032 |
| 6 | 8.673972603 | −4.39982 | 27.58888889 | −28.3714 |
| 7 | 11.28528529 | −1.7885 | 57.59722222 | 1.636902 |
| total | 13.07378775 | | 55.96031989 | |

publications and relative research interest both showed a downward trend, which suggests that the researchers' interest have shifted and more innovation needs to be explored in the pancreatic stents. In addition, we also focused on the countries and author of research outputs. The United States, Japan and Germany remain to be the countries with the largest number of publications on pancreatic stents. The results indicated the developed countries occupied main position in the field. After measuring the top 29 authors on pancreatic stents, we made the author relationship network. The authors in the field have close cooperation, emphasizing the importance of cooperation. By paying attention to these authors, we would have a general understanding of the research direction and hotspots in this field. In order to further track research trends, journals are also the focus of attention. Therefore, we measured the most active journals, considering as the central journals in the relevant fields such as Gastrointestinal endoscopy, Endoscopy, World journal of

**Table 5  The cluster analysis of 8 clusters.**

| Cluster | Number of MeSH terms[a] | Cluster analysis |
|---|---|---|
| 0 | 23,34,56,58,69,72,79,82 | Stents placement in pancreatic neoplasms |
| 1 | 5,9,10,19,21,24,27,36,39,40,41 | The complications of stents placement in bile duct neoplasms and pancreatic neoplasms |
| 2 | 8,35,45,49,50,52,54,71,73,74,75,76 | postoperative complications after stent placement such as pancreaticoduodenectomy |
| 3 | 1,2,3,6,11,15,17,18,46,47,51,60,67,80 | Stents for the prevention of pancreatic fistula following pancreaticoduodenectomy |
| 4 | 12,13,14,22,28,37,48,61,65 | https://www.ncbi.nlm.nih.gov/pubmed/22185981 pancreatic duct stent can reduce the incidence of post-ERCP pancreatitis (PEP) |
| 5 | 4,25,29,31,32,33,38,42,57,77 | The diagnosis, surgery and therapy of pancreatitis |
| 6 | 16,43,55,59,62,64,68,78,81,83 | Pancreatic ducts changes in patients with chronic pancreatitis |
| 7 | 7,21,26,30,44,53,63,66,70 | Stent placement in endoscopic pancreatic pseudocyst drainage |

**Notes.**

[a]Represents the serial number of high-frequency MeSH terms.

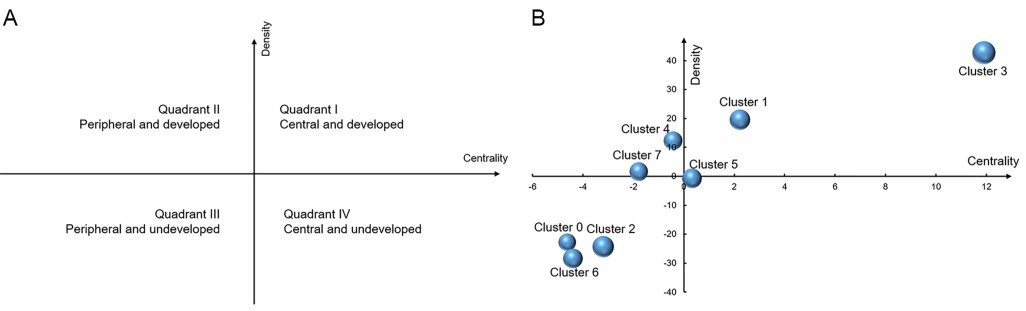

**Figure 4  Strategic diagram for the application of stents in pancreatic diseases.** (A) The meaning of strategic diagram. (B) The strategic diagram of the 8 clusters for the application of stents in pancreatic diseases.

gastroenterology. The high-frequency MeSH terms may reflect the research hot spots. The 83 high-frequency major MeSH terms were achieved by the co-occurrence in the same paper, which represented the research content in the field. Yearly distribution trends on different MeSH terms had the same fluctuating trend.

Eighty-three hot major MeSH terms were clustered into eight clusters. The network revealed that these MeSH terms existed complex relationship network. Endoscopic retrograde ERCP in acute and chronic pancreatitis and imaging methods as an auxiliary method of stent placement are located in the second quadrant. Cluster 1 and 3 are located in the first quadrant, including the complications of stent placement in bile duct neoplasms and pancreatic neoplasms and stents for the prevention of pancreatic fistula following pancreaticoduodenectomy. The two topics are current research center and hot topics for pancreatic stents. And cluster 0, 2, 6 are located in the third quadrant, which suggesting

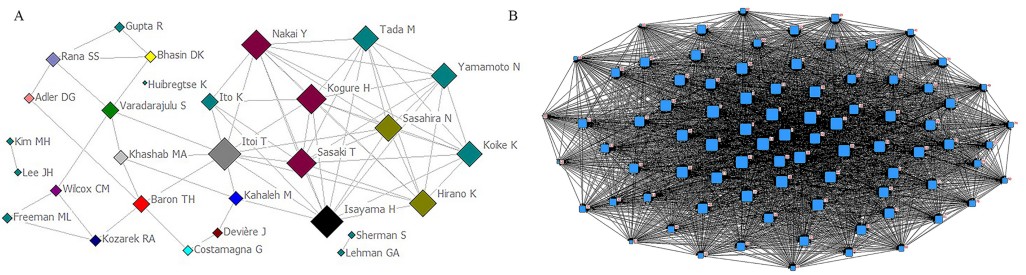

**Figure 5** **Social network analysis.** (A) The top 29 author relationship network. The size and location of nodes represent the centrality of an author in the social network. (B) The network of high-frequent major MeSH terms. Nodes suggest high-frequent major MeSH terms. The size and location of nodes represent the centrality of a MeSH term in the network structure map. Links stand for the connection between MeSH terms, and the number or thickness of the lines stands for the co-occurrence frequency of high-frequent major MeSH terms.

that the three topics are at the margin and not yet mature, including stents placement in pancreatic neoplasms, the postoperative complications after stent placement therapy such as pancreaticoduodenectomy and pancreatic ducts changes for patients in chronic pancreatitis. In the meanwhile, complications such as pancreatitis associated with stent implantation could have potential research value in the fourth quadrant, which are the research center, however, not yet mature. Therefore, the topic could become potential hotspots in the future science research. Then the 8 topics would be introduced respectively.

## Stents placement in pancreatic neoplasms

Increasing numbers of patients with resectable pancreatic neoplasms are receiving neoadjuvant therapy such as stents placement. Tumor growth in pancreatic neoplasms often leads to invasion of other organs and biliary obstruction, resulting in repeated stent placement (*Shi et al., 2019*). The self-expandable metal stents possess effectiveness and safety in achieving durable biliary drainage for patients with pancreatic neoplasms (*Aadam et al., 2012*; *Van der Horst et al., 2014*). For example, the covered self-expanding metal stents is used for the therapy of biliary tract hemorrhage induced by advanced pancreatic cancer-induced portal biliary disease (*Kim et al., 2016*).

## The complications of stent placement in bile duct neoplasms and pancreatic neoplasms

As for pancreatic neoplasms, preoperative biliary drainage (PBD) promotes complications compared with surgery without PBD. The result could be associated with the plastic stents utilized. However, metal stents might decrease the PBD-associated complications (*Tol et al., 2016*). It has been confirmed that biliary stents could remarkably increase liver volume in both hilar and distal bile duct neoplasms (*Lee et al., 2014*). Endoscopic retrograde biliary drainage of metal bile duct stents are widely used for biliary obstruction. The application of bile duct stents has also led to an increasing number of complications. The main complications of pancreatic stents include migration, stent occlusion, and pancreatic ductal changes (*ASGE Technology Assessment Committee et al., 2013*).

## Postoperative complications after stent placement such as pancreaticoduodenectomy

Pancreatic fistula is a leading complication following pancreaticoduodenectomy. *Pessaux et al. (2011)* have reported that external pancreatic duct stent reduces pancreatic fistula rate following pancreaticoduodenectomy. Obstructive jaundice is one of the known risk factors for treatment failure following hepatectomy for patients with hilar cholangiocarcinoma. In palliative care, self-expanding metal stents have a rapid reduction in bile duct pressure and reduce complication rates, while providing patients with adequate and rapid biliary drainage (*Grünhagen et al., 2013*).

## Stents for the prevention of pancreatic fistula following pancreaticoduodenectomy

It is necessary to prevent pancreatic fistula after pancreaticoduodenectomy in stent placement. The incidence of pancreatic fistula in patients undergoing pancreaticoduodenectomy is as high as 56% and is considered to be a main factor on morbidity and mortality in patients following pancreaticoduodenectomy (*Dong et al., 2016*; *Brown et al., 2014*). And external duct stents placement could reduce the occurrence for clinically relevant postoperative pancreatic fistula (*Motoi et al., 2012*).

## Prophylactic pancreatic duct stent can reduce the incidence of post-ERCP pancreatitis (PEP) and complications such as pancreatitis associated with stent implantation

Endoscopic retrograde ERCP was first introduced in 1968. As a diagnostic tool, it was used to assess the disorders of pancreas (*Riff & Chandrasekhara, 2016*). As a most common complication of ERCP, the incidence of PEP is still as high as 15% in high-risk cases (*Elmunzer, 2017*). A small number of patients could develop severe pancreatitis. Pancreatitis is a common and serious complication for endoscopic retrograde ERCP. Prevention of pancreatitis after ERCP remains the focus of clinical and research. Relevant strategies could decrease the occurrence of post-ERCP pancreatitis including patient selection, risk stratification, surgical techniques, pancreatic stenting, and drug prophylaxis. Placement of the pancreatic stent is a relatively new and increasingly popular method of reducing the risk of pancreatitis after ERCP (*Shi et al., 2014*). Prophylactic pancreatic stent placement decreases the incidence of pancreatitis after ERCP in high risk patients and reduces the severity of this condition (*Freeman, 2007*). In summary, placement for pancreatic duct stent decreases the incidence of pancreatitis (*Sofuni et al., 2011*).

## Pancreatic duct changes in patients with chronic pancreatitis

It is essential to prevent pancreatic duct changes such as pancreatic leakage or pancreatic duct patency after pancreaticoduodenectomy. In duct-to-mucosa anastomosis, placement of the stent could be an effective mean of dilating the pancreatic duct (*Téllez-Aviña et al., 2018*). Pancreatic stent is used to improve painful, obstructive chronic pancreatitis (*Samuelson et al., 2016*).

### Stent placement in endoscopic pancreatic pseudocyst drainage

Pancreatic pseudocyst is one of the common local complications of acute and chronic pancreatitis. And endoscopic pancreatic pseudocyst drainage has been widely applied in the treatment of pancreatic pseudocysts (*Madder et al., 2016*). Endoscopic drainage has the advantages of small invasiveness, short recovery time, low cost and low complication rate (*Shah et al., 2015*), like interventional endoscopic ultrasonography has been increasingly used to manage pseudocyst formation (*Vilmann et al., 2015*). As an example, *Varadarajulu et al. (2013)* have found that, compared with surgical bladder anastomosis, patients with endoscopy pancreatic pseudocyst drainage experience rarely recurrence of pseudocyst during follow-up.

## CONCLUSION

We analyzed the literature on pancreatic stents based on bibliometric analysis. Finally, 83 high-frequent MeSH terms and eight topics were found. And we found how to reduce the incidence of postoperative complications and improve the prognosis of patients with pancreatic diseases by stent implantation is still the focus of future research. This conclusion could provide potential and invaluable insight for researchers in the further research.

**Abbreviations**

| | |
|---|---|
| **MeSH** | Medical Subject Headings |
| **BICOMB** | Bibliographic Item Co-Occurrence Matrix Builder |
| **PMIDs** | PubMed Unique Identifiers |
| **PBD** | preoperative biliary drainage |
| **ERCP** | cholangiopancreatography |

### Funding
The authors received no funding for this work.

### Competing Interests
The authors declare there are no competing interests.

### Author Contributions
- Xuan Zhu performed the experiments, analyzed the data.
- Xing Niu performed the experiments, analyzed the data, authored or reviewed drafts of the paper.
- Tao Li contributed reagents/materials/analysis tools, prepared figures and/or tables.
- Chang Liu prepared figures and/or tables.
- Lijie Chen authored or reviewed drafts of the paper.
- Guang Tan conceived and designed the experiments, approved the final draft.

## Data Availability

The data is based on literature and other publications. All publications came from PubMed. The raw measurements are available in the Supplemental Files.

## Supplemental Information

Supplemental information for this article can be found online at http://dx.doi.org/10.7717/peerj.7674#supplemental-information.

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
