# Peer review of "Identification of research trends concerning application of stent implantation in the treatment of pancreatic diseases by quantitative and biclustering analysis: a bibliometric analysis"

_PeerJ, doi:10.7717/peerj.7674_

## Round 0.1 · original submission · Major Revisions

The manuscript received detailed remarks from 3 reviewers. The problem is important and the method of bibliometic analysis is quite novel and fits well to the journal scope. But statistical adjustment of the results should be improved along with all the comments raised by the reviewers.
I encourage resubmit the manuscript to PeerJ after the revision.

Reviewer 1 ·

Basic reporting

1) The authors are using the 'Mature Research' term for the characterization of their clusters. Is it a commonly used term? In case if it is so, it is necessary to provide references on some other works related to its application. Otherwise, it is needed to substantiate in more detail the possibility of using this criterion to classify research areas according to their maturity.

2) In the paper on the lines 157-159, the authors are saying: "The first article was published in 1977. As time went by, the volume of publications increased year by year. By 2015, it had a downward trend." At the same time, the conclusion in the abstract is: "In our study, we found 8 topics concerning the application of stent implantation in the treatment of pancreatic diseases. And how to reduce the incidence of postoperative complications and improve the prognosis of patients with pancreatic diseases by stent implantation could become potential hotspots in the future research."

How can the authors comment that despite the falling interest of researchers to the "application of stents in pancreatic diseases," the identified research directions in this area can be considered as potential hotspots in future?

3) It is necessary to provide a more detailed description in the article of how exactly the information, obtained based on the analysis of the reconstructed social network of authors, was used in the cluster analysis to assess the maturity of research areas and identification of the hot spots.

3) The list of provided Mesh terms is unreadable in Figures 2 and 3.

4) The numbering of clusters is missing in Figure 4.

5) The English language should be improved.

Experimental design

no comment

Validity of the findings

no comment

Reviewer 2 ·

Basic reporting

The manuscript provides information about the research hotspots in the area of stent implantation in the management of pancreatic disorders based on relevant PubMed publications using a bibliometric analysis by co-word analysis with visualization and strategic diagram of the main areas of interest. The article is original and acceptable in its current form after minor changes in respect to gramma. No plagiarism was observed and no part of this paper was published elsewhere. The abstract summarizes well the content of the article.The reference list is sufficient and contains 43 papers.

Experimental design

The introduction reveals the problem with pancreatic stent implantation and the usefulness of bibliometric methods and co-word analysis. The aims of the paper are clear and meet the discussion of the body text. The research methods are well explained.

Validity of the findings

The results and discussion are carried out in a satisfactory manner with good visualization.
Using Bibliographic Item Co-Occurrence Matrix Builder and biclustering analysis, the authors revealed the countries and journals, contributing mostly for the research investigations through largest number of publications in the discussed area. The authors managed to find 8 topics concerning the application of stent implantation in the treatment of pancreatic diseases, of which the complications of stent placement in bile duct neoplasms and pancreatic neoplasms and stents for the prevention of pancreatic fistula following pancreaticoduodenectomy are current research center and hot topics for pancreatic stents. The conclusion supports the presented research.

Additional comments

The full name of the abbreviation MeSH (Medical Subject Headings) should appear when it is first mentioned in the abstract and line 102.
Two of the discussed areas of interest according to post-ERCP pancreatitis and stent’s complication, namely ''Prophylactic pancreatic duct stent can reduce the incidence of post-ERCP pancreatitis (PEP)'' and ''Complications such as pancreatitis associated with stent implantation'' consist of overlapping information. However, those two focuses might become one.
As the statement (Line 346) that “patients with endoscopy pancreatic pseudocyst drainage had no recurrence of pseudocyst during follow-up” is observed by Varadarajulu et al., it would be more appropriate either to mentioned the name of the author within the text, or to conclude that patients with endoscopy pancreatic pseudocyst drainage experience rarely recurrence of pseudocyst during follow-up.
Spelling might be corrected on few locations:
Line 107: Publication trends were (instead of “was” as it is in the maniscript)
Line 110: was an original article (instead of articles)
Line 160: [MeSH] instead of [mesh]
Line 232: Cluster 5 is (instead of are)
Line 233: indicating that it has (instead of "they have")
Line 251: the volume of relevant publications was (instead of ''were'')
Line 254: suggests that (instead of suggesting)
Line 256: The United States, Japan and Germany remain (instead of ''remains'')
Line 264: Gastrointestinal endoscopy (instead of ''gastrointestinal'')
Line 289: drainage for patients with pancreatic neoplasms (instead of ''in pancreatic neoplasms'')
Line 306: for patients with hilar cholangiocarcinoma (instead of ''patients in hilar..'')
Line 320: As a most common complication (instead of ''complications'')
Line 324: after ERCP in high risk patients (instead of ''after ERCP patients in high risk'')
Line 337: an effective mean (instead of ''an effective means'')

Reviewer 3 ·

Basic reporting

- Language should be improved. There are some grammatical errors and typos, i.e.:
+ ... Publications on the application of stents in pancreatic diseases was retrieved from PubMed ...
+ ... the covered metal stents reduces the incidence of ...
+ ... If a paper concerning application of stents in pancreatic diseases was an original articles, ...
+ ... Publication trends was retrieved from GoPubMed ...
+ ...
- References are weak. There is a need to provide and discuss more related works.
- Figures are not clear and transparent. For example, I cannot read the text in Fig. 1, 2, 3.
- The authors should provide more description in their figure's legends.
- In Fig. 5, the network is confusing. I cannot catch the idea here. It needs to be shown clearly.
- The authors should use footnotes to explain the values in their tables.

Experimental design

- The authors retrieved data on May 15th, 2018, however, what is the idea of the field "2018/03/26" [PDAT]?
- To mention the PubMed database, the authors should say that it has been used to retrieve data in some of the biomedical research, such as https://doi.org/10.1016/j.cmpb.2019.05.016 and https://doi.org/10.1007/s00438-019-01570-y.
- "MeSH" abbreviation has not been defined in the first use.

Validity of the findings

- In Table 1, what are the difference between the values in "Percentage, %" column and the percentage in "Frequency n (%)" column? If not, it is redundancy.
- One of the most concerns is the low distributions of the values. It makes this paper become less contribution and cannot conclude a lot of things. For example, in Table 1, 2, 3, the percentage values of <1% cannot be considered as high frequency. The authors listed some of the topmost values, but they are still too low to make a difference.
- How to select the significant level of the results in Table 1, 2, 3? For example, how did the authors select the cut-off point in these tables? There is a need to have a statistical test for that, i.e. to show the p-value, which has been applied in the previous publications: https://doi.org/10.1186/s12859-016-1163-x and https://doi.org/10.1142/S0219720019500057.
- Table 4 is useless. It did not say anything because everything has been mentioned in the previous table.

---

## Round 0.2 · accepted · Accept

The reviewers had no more remarks

Reviewer 3 ·

Basic reporting

No comment

Experimental design

No comment

Validity of the findings

No comment

Additional comments

My previous comments have been addressed.